# Triethylamine inhibits influenza A virus infection and growth via mechanisms independent of viral neuraminidase and RNA-dependent RNA polymerase

Masaki Shoji[1]*, Kensuke Nakaoka[1], Momiji Ishikawa[1], Yusuke Kasai[2], Tomoyuki Esumi[3], Etsuhisa Takahashi[4], Hiroshi Kido[4], Hiroshi Imawaga[2], Takashi Kuzuhara[1]*

1 Laboratory of Biochemistry, Faculty of Pharmaceutical Sciences, Tokushima Bunri University, Tokushima, Japan, 2 Chemistry of Functional Molecules, Faculty of Pharmaceutical Sciences, Tokushima Bunri University, Tokushima, Japan, 3 Laboratory of Medicinal Chemistry, Faculty of Pharmaceutical Sciences at Kagawa, Tokushima Bunri University, Kagawa, Japan, 4 Division of Enzyme Chemistry, Institute for Enzyme Research, Tokushima University, Tokushima, Japan

* masaki-shoji@ph.bunri-u.ac.jp (MS); kuzuhara@ph.bunri-u.ac.jp (TK)

## Abstract

Triethylamine ($Et_3N$) is a proton ($H^+$) acceptor that is widely used in various industrial organic synthesis processes, including the production of pharmaceuticals, agrochemicals, and polymers. Inhalation of high $Et_3N$ concentrations can damage human respiratory tract and lungs. Given the compound's known reactivity and membrane-penetrating properties, we hypothesized that non-toxic concentrations of $Et_3N$ may exert modulatory effects on virus–host interactions in epithelial cells. We thus investigated the anti-influenza activity of $Et_3N$ and found that it enhanced the viability of influenza A H1N1 and H3N2 virus-infected Madin–Darby canine kidney (MDCK) cells. Non-cytotoxic $Et_3N$ concentrations reduced the number of infected cells and suppressed influenza A virus nucleoprotein expression as well as viral gene and antiviral host gene upregulation in infected MDCK cells. Selectivity index values of $Et_3N$ against influenza A virus infection, ranging from approximately 10 to over 50. These findings indicated that $Et_3N$ inhibited influenza A H1N1 and H3N2 viral infections. Additionally, $Et_3N$ suppressed influenza A H1N1 and H3N2 virus titers in the infected MDCK cell culture supernatant, suggesting that it inhibited viral growth in infected cells. This implies that $Et_3N$ may suppress influenza A virus release and/or replication by targeting viral or host cell factors. However, $Et_3N$ did not inhibit influenza A viral neuraminidase or RNA-dependent RNA polymerase activity, which are involved in viral release and replication, respectively. These results suggest that $Et_3N$ targets other viral proteins or host cell factors essential for influenza A virus growth. Our findings demonstrate that $Et_3N$ exerts anti-influenza activity, providing new insights into the development of antiviral agents based on $Et_3N$ skeleton.

**Data availability statement:** All relevant data are within the manuscript and its Supporting Information files.

**Funding:** This work was supported by grants from Tokushima Bunri University and Tokushima Bunri University for Educational Reform and Collaborative Research (No. TBU2024-2-2) (to MS). This work was supported in part by a Grant from Japan Society for the Promotion of Science (JSPS) KAKENHI (23K06510) (to MS). The funders had no role in the study design, data collection and analysis, decision to publish, or manuscript preparation.

**Competing interests:** The authors have declared that no competing interests exist.

## Introduction

Seasonal influenza remains a significant global public health concern, causing recurrent outbreaks that lead to substantial morbidity and mortality. Currently available antiviral agents include neuraminidase (NA) inhibitors, which block the release of progeny virions, and M2 ion channel blockers, which interfere with viral uncoating [1]. However, increasing reports of viral resistance to these drugs have raised serious concerns regarding their long-term efficacy [2]. In addition, recently developed antivirals targeting viral RNA polymerase-related enzymes have also encountered resistance-associated mutations [2–5]. These challenges underscore the urgent need for novel antiviral compounds that act through alternative mechanisms of action.

Triethylamine ($Et_3N$) is a tertiary amine and a strong proton ($H^+$) acceptor capable of forming hydrogen bonds. It is a volatile, colorless liquid with a distinct odor and widely used as a catalyst, acid scavenger, preservative, and raw material in various industrial organic synthesis processes, including pharmaceutical, agrochemical, and polymer production [6–9]. However, inhalation of high $Et_3N$ concentrations can cause severe irritation to the human respiratory tract and lungs [6]. $Et_3N$ inhibits monoamine oxidase activity in the liver and brain of mice both *in vitro* and *in vivo*, leading to central nervous system stimulation [10]. It also inhibits sulfotransferase activity toward androsterone and dehydroepiandrosterone, but not cortisol or 2-naphthol, in rat hepatic cytosolic preparations [11,12]. Additionally, $Et_3N$ inhibits protein degradation, induces lysosomal swelling (vacuolation), and suppresses protein synthesis in rat hepatocytes under nutrient-free conditions [13]. Akesson et al. examined the pharmacokinetics of $Et_3N$ and its metabolite, $Et_3N$-*N*-oxide ($Et_3NO$), in four human participants after oral and intravenous administration. The study revealed that $Et_3N$ is efficiently absorbed from the gastrointestinal tract and partially metabolized into $Et_3NO$ [14]. Both $Et_3N$ and $Et_3NO$ are excreted in urine, with half-lives of approximately 3–4 h in the plasma and urine [14]. Although $Et_3N$ has been investigated in various contexts, its biological and pharmacological effects of antiviral activity at non-toxic concentrations on infectious diseases of the respiratory tract have not been extensively studied. Given the compound's known reactivity and membrane-penetrating properties, we hypothesized that $Et_3N$ may exert modulatory effects on virus–host interactions in epithelial cells. We thus speculated that $Et_3N$ might influence influenza A virus infection and/or growth efficiency.

In this study, we investigated the anti-influenza activity of $Et_3N$ and found that it inhibited influenza A H1N1 and H3N2 virus infections in Madin–Darby canine kidney (MDCK) cells at non-cytotoxic concentrations when applied concurrently with the virus. In addition, $Et_3N$ inhibited viral growth in infected cells. However, it did not inhibit influenza A viral NA or RNA-dependent RNA polymerase (RdRp) activities, which are involved in viral release and replication. These findings suggest that $Et_3N$ targets other viral proteins or host cell factors essential for influenza A virus growth. Our results demonstrate that $Et_3N$ exerts anti-influenza activity, indicating new uses of low concentration of $Et_3N$ and providing new insights into the development of antiviral agents based on $Et_3N$ skeleton.

## Materials and methods

### Compound preparation

$Et_3N$ (CAS RN: 121-44-8, MW = 101.19) was purchased from Tokyo Chemical Industry Co. Ltd. (Tokyo, Japan). (+)-(S)-Bakuchiol (baku) that have the anti-influenza activity was synthesized using as reported previously [15,16]. Oseltamivir carboxylate (OC; Ro 64–0802), a known NA inhibitor [17], was prepared as described previously [17]. Ribavirin, a viral genome replication inhibitor [18], was purchased from MP Biomedicals (Illkirch, France). Liquid $Et_3N$ (>99.0%) was diluted 71.25 times with sterilized Milli-Q water (Water; filtered through a 0.22 μm filter) to prepare a 100 mM (>1.39%) stock solution. Stock solutions of baku (10 mM), OC (1 mM), and ribavirin (10 mM) were prepared by separately dissolving them in dimethyl sulfoxide (DMSO).

### MDCK cells

MDCK cells were cultured in the growth medium (high-glucose Dulbecco's modified Eagle's medium [DMEM] with L-glutamine [Nacalai Tesque, Kyoto, Japan] supplemented with 10% fetal bovine serum [FBS; Thermo Fisher Scientific, MA, USA], and 100 unit/mL penicillin and 100 μg/mL streptomycin [Thermo Fisher Scientific]) and maintained at 37 °C in a 5% $CO_2$ atmosphere.

### Influenza A virus strains

The A/human/Puerto Rico/8/1934 (A/PR/8/34; H1N1), A/human/WSN/1933 (A/WSN/33; H1N1), and A/human/Aichi/2/1968 (A/Aichi/2/68; H3N2) strains of influenza A virus, provided by Takahashi and Kido [19], were used in the experiments. These viruses were propagated in MDCK cells using the infection medium (high-glucose DMEM with L-glutamine [Nacalai Tesque] supplemented with 1% bovine serum albumin [BSA; FUJIFILM Wako Pure Chemical, Osaka, Japan], 100 unit/mL penicillin and 100 μg/mL streptomycin [Thermo Fisher Scientific]) supplemented with 3 μg/mL L-tosylamido-2-phenyl ethyl chloromethyl ketone (TPCK)-treated trypsin (Sigma-Aldrich, MO, USA). Viral titers (plaque-forming units per milliliter [PFU/mL]) were determined by immunostaining for influenza A viral nucleoprotein (NP), as previously described [20,21].

### WST-8 assay to assess $Et_3N$-treated MDCK cell viability

MDCK cells were cultured in a 96-well plate ($1 \times 10^4$ cells/well) and treated with $Et_3N$ dissolved in Water at various concentrations ($Et_3N$ [mM]:Water [%] = 10 mM:10%; 5 mM:5%; 2 mM:2%; 1 mM:1%; 0.5 mM:0.5%; 0.25 mM:0.25%) in growth medium. The cells were then incubated for 72 h at 37 °C in a 5% $CO_2$ atmosphere. Water (0.25–10%) was used as the negative control. After incubation, cell viability was assessed using a WST-8 assay with Cell Count Reagent SF (Nacalai Tesque) following the manufacturer's instructions.

### Naphthol blue-black (NB) staining

To assess influenza virus-infected MDCK cell viability, NB staining assay was performed as previously described [20,21]. Briefly, $Et_3N$ (7.8–1000 μM) was mixed either without virus (−) or with influenza virus at a multiplicity of infection (MOI) of 0.5 for A/PR/8/34, 0.5 for A/WSN/33, or 10 for A/Aichi/2/68 in growth medium, and incubated for 30 minutes at 37 °C in 5% $CO_2$. Water (0.0078–1%) and baku (0.2–25 μM) were used as negative and positive controls, respectively. The resulting mixtures were added to MDCK cells and incubated for 72 h at 37 °C in 5% $CO_2$. Thereafter, cells were stained with NB to evaluate viability.

### Immunofluorescence (IF) staining of influenza A virus-infected MDCK cells

To assess the number of influenza virus-infected MDCK cells, IF staining targeting the viral NP was performed as previously described [20,21]. Briefly, $Et_3N$ (125–1000 μM) was mixed with influenza viruses at a MOI of 0.01 for A/PR/8/34, 0.01 for A/WSN/33, or 0.1 for A/Aichi/2/68 in growth medium. Water (1%) and baku (10 μM) were used as

negative and positive controls, respectively. The resulting mixtures were added to MDCK cells into 96-well plates ($1 \times 10^4$ cells/well) and incubated for 24 h at 37 °C in 5% $CO_2$. Cells were fixed with 4% paraformaldehyde and permeabilized with 0.3% Triton X-100. Primary and secondary antibodies used were mouse anti-influenza A NP monoclonal antibody (FluA-NP 4F1; Southern Biotech, AL, USA) and Alexa Fluor 488-conjugated goat anti-mouse IgG (H + L) (Thermo Fisher Scientific), respectively. Cell nuclei were stained with 4',6-diamidino-2-phenylindole (DAPI; Thermo Fisher Scientific). The stained wells were imaged using a fluorescence microscope (BIOREVO BZ-X700; Keyence, Osaka, Japan), and the percentage of NP-positive cells was calculated relative to the total number of DAPI-stained nuclei.

### Western blotting (WB) analysis

To assess the expression of influenza viral NP in virus-infected MDCK cells, WB analysis performed as previously described [21]. Briefly, $Et_3N$ (125–1000 μM) was mixed with influenza viruses at a MOI of 0.01 for A/PR/8/34, 0.01 for A/WSN/33, or 0.1 for A/Aichi/2/68 in growth medium. DMSO (1%) and baku (10 μM) were used as negative and positive controls, respectively. The resulting mixture was added to cells into 96-well plates ($1 \times 10^4$ cells/well) and incubated for 24 h at 37 °C in 5% $CO_2$. Cells were then lysed with sample buffer containing 62.5 mM Tris-HCl (pH 6.8), 2% sodium dodecyl sulfate (SDS), 10% glycerol, 0.005% bromophenol blue, and 5% β-mercaptoethanol. The lysates were boiled for 5 minutes and subjected to SDS–polyacrylamide gel electrophoresis. Separated proteins were transferred to polyvinylidene fluoride membranes (Millipore, Billerica, MA, USA). The primary antibodies used were a mouse anti-influenza A NP monoclonal antibody (FluA-NP 4F1; Southern Biotech) and a rabbit anti-β-ACTIN monoclonal antibody (13E5; Cell Signaling Technology, MA, USA). The corresponding secondary antibodies were a horseradish peroxidase (HRP)-conjugated goat anti-mouse IgG1 (Southern Biotech) and an HRP-conjugated goat anti-rabbit IgG (Sigma-Aldrich). Protein signal intensities were quantified using ImageJ software, and NP expression levels were normalized to those of β-ACTIN.

### The selectivity index (SI) calculation

The SI values were calculated based on IF staining or WB analysis of influenza NP expression in $Et_3N$-treated MDCK cells infected with influenza A virus strains [22]. SI was determined using the formula: SI = 50% cytotoxic concentration ($CC_{50}$)/ 50% inhibitory concentration ($IC_{50}$). The $CC_{50}$ of $Et_3N$ in MDCK cells was obtained from WST-8 assay results. The $IC_{50}$ values of $Et_3N$ against A/PR/8/34, A/WSN/33, and A/Aichi/2/68 strains were calculated from the IF staining and WB data. All $CC_{50}$ and $IC_{50}$ values were calculated using GraphPad Prism version 10 (GraphPad Software, CA, USA). An SI greater than 1.0 indicates that $Et_3N$ exhibits greater antiviral efficacy than cytotoxicity in MDCK cells.

### Reverse transcription and quantitative polymerase chain reaction (RT-qPCR) analysis

To assay influenza A virus and antiviral host gene expression in virus-infected MDCK cells, RT-PCR analysis performed as previously described [21]. Briefly, 1000 μM $Et_3N$ was mixed either without virus (−) or with influenza viruses at a MOI of 0.01 for A/PR/8/34 in growth medium. Water (1%) and baku (10 μM) were used as negative and positive controls, respectively. The resulting mixtures were added to MDCK cells into 96-well plates ($1 \times 10^4$ cells/well) and incubated for 24 h at 37 °C in 5% $CO_2$. Total RNA was extracted from the cell lysates using a RNeasy Mini Kit (Qiagen GmbH, Germany). cDNA was synthesized from total RNA using SuperScript VILO (Thermo Fisher Scientific), according to the manufacturer's instructions. The synthesized cDNA was used as a template for qPCR performed using THUNDERBIRD Next SYBR qPCR Mix (TOYOBO, Osaka, Japan). The primers used are listed in Table S1. PCR and data analyses were conducted using an Applied Biosystems QuantStudio 3 Real-Time PCR System (Thermo Fisher Scientific). Relative expression levels were calculated using the ΔΔCT method.

### Influenza A viral growth assay

To assess influenza virus growth, Viral titers were measured in influenza A virus-infected MDCK cell culture supernatant as previously described [20,21]. MDCK cells were infected with influenza viruses at a MOI of 0.0001 MOI for A/PR/8/34, 0.0001 for A/WSN/33, or 0.001 for A/Aichi/2/68 in infection medium for 1 h at 37 °C with 5% $CO_2$, before being treated with compounds (pre-infection experiment). The virus-infected cells were treated with 2 mM $Et_3N$ in infection medium supplemented with 3 µg/mL TPCK-treated trypsin (Sigma-Aldrich). Water (2%) and baku (20 µM) were used as negative and positive controls, respectively. After incubating 48 h at 37 °C with 5% $CO_2$, viral titers in the culture supernatant were determined (PFU/mL).

### NA inhibition (NAI) assay using influenza A viral particles

NAI assay using influenza viruses was performed as previously described [20]. Briefly, 2 mM $Et_3N$ was diluted in the NA assay buffer in a 96-well black plate (Thermo Fisher Scientific). Water (2%) and OC (0.5 µM) were used as negative and positive controls, respectively. The wells were added to either without virus (–) or with influenza viruses at a PFU of $1 \times 10^2$ for A/PR/8/34, $5 \times 10^2$ for A/WSN/33, or $1 \times 10^4$ for A/Aichi/2/68 in NA assay buffer and then incubated for 30 min at 37 °C with 5% $CO_2$. Subsequently, each sample was mixed with 12.5 µM 2′-(4-methylumbelliferyl)-α-D-N-acetylneuraminic acid (Sigma-Aldrich). After incubation at 37 °C with 5% $CO_2$ for 0, 3, 6, or 24 h, the fluorescence intensity was measured at 365 nm excitation and 445 nm emission using an Infinite 200 PRO M Plex (Tecan, Zurich, Switzerland).

### Influenza A viral minigenome assay

A minigenome assay based on the *Firefly* and *Renilla* luciferase (LUC) systems was performed as previously described [20,23]. MDCK cells in a 96-well plate ($1 \times 10^4$ cells/well) were transfected with the plasmids for influenza viral minigenome assay via reverse transfection following the manufacturer's instructions. The cells were treated with 2 mM $Et_3N$ and incubated at 37 °C with 5% $CO_2$ 6 h post-transfection. Water (2%) and ribavirin (25 µM) were used as negative and positive controls, respectively. After 24 h of incubation, *Firefly*_LUC and *Renilla*_LUC activities in the transfected cells were measured using the Dual-Glo LUC assay system (Promega). The LUC activities were expressed as relative light units (RLU), and the ratio of *Firefly*_LUC to *Renilla*_LUC (*Firefly*_LUC/*Renilla*_LUC) activity was calculated.

### Statistical analysis

All results are presented as the mean ± standard error of the mean (SEM). The statistical significance of comparisons between more than two groups was analyzed using one-way analysis of variance (ANOVA), followed by Dunnett's or Tukey's post-hoc tests. Results were considered statistically significant at $P < 0.05$.

## Results

### $Et_3N$ enhanced influenza A H1N1 and H3N2 virus-infected MDCK cell survival

We assessed the cytotoxicity of $Et_3N$ (Fig 1A) in MDCK cells using the WST-8 assay. The viability of 10 mM $Et_3N$-treated cells reduced after 72 h of incubation than that of only Water-treated cells (Fig 1B). However, ≤ 5 mM $Et_3N$-treated cells no reduction in viability (Fig 1B). These results indicate that exposure to ≤5 mM $Et_3N$ for 72 h did not induce cytotoxicity in MDCK cells.

Subsequently, we examined the effects of $Et_3N$ on influenza A H1N1 (A/PR/8/34 and A/WSN/33) or H3N2 (A/Aichi/2/68) viruses-infected MDCK cell survival using NB staining. Cells treated with all tested $Et_3N$ concentrations stained blue in the absence of viral infection (Fig 1C), confirming that these concentrations were non-cytotoxic. The cells infected with viruses and exposed to Water were not stained (Fig 1C), indicating a loss of viability. In contrast, the cells infected with A/PR/8/34 or A/WSN/33 and treated with 1.56–25 µM baku stained blue (Fig 1C), demonstrating a protective effect.

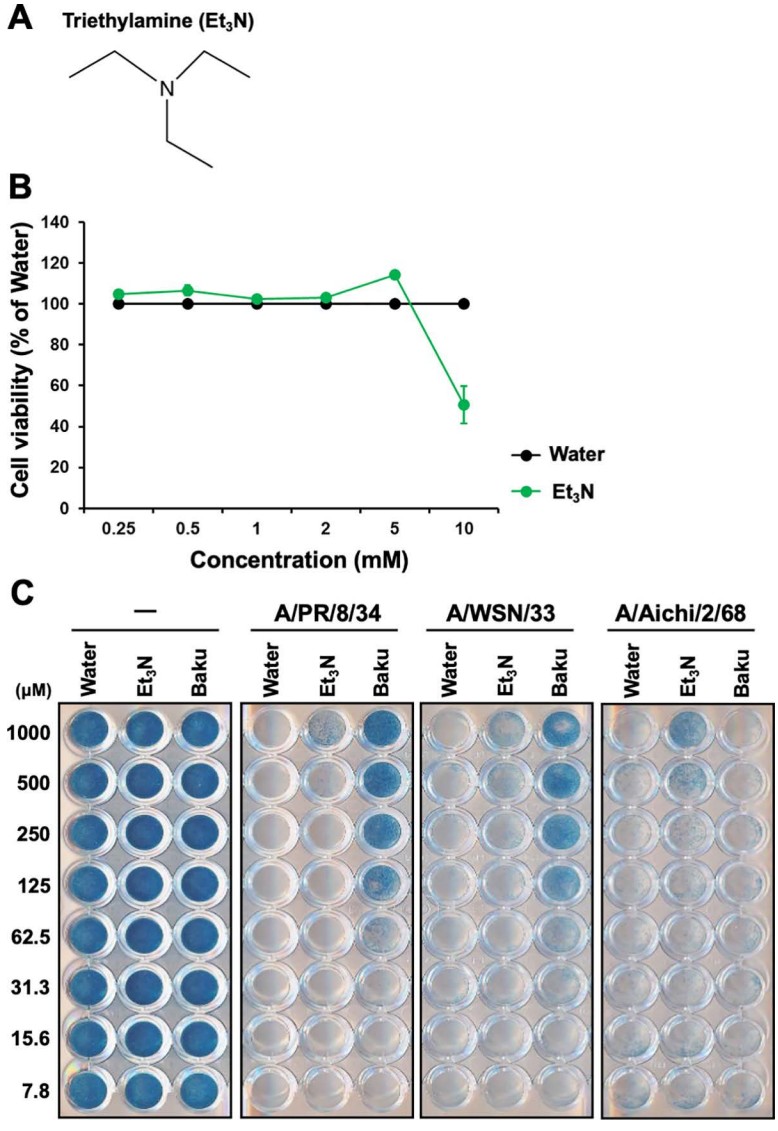

**Fig 1. Effects of Et₃N on survival in influenza A virus-infected MCDK cells. (A)** Chemical structure of Et₃N. **(B)** The cytotoxicity of Et₃N in MDCK cells using the WST-8 assay. Data represent the mean ± SEM (n = 8) of two independent experiments. **(C)** The effects of Et₃N on influenza A H1N1 (A/PR/8/34 or AWSN/33) and H3N2 (A/Aichi/2/68) virus-infected MDCK cell viability using NB staining. Data are representative of four independent experiments, with results consistently reproducible across these experiments.

Similarly, treatment with 500–1000 µM Et₃N resulted in blue staining of cells infected with A/PR/8/34, A/WSN/33, or A/Aichi/2/68 virus (Fig 1C), indicating that the cells remained viable despite viral infection. These results demonstrated that Et₃N enhanced influenza A H1N1 and H3N2 virus-infected MDCK cell viability.

### Et₃N reduced the number of influenza A virus-infected cells

We measured the number of virus-infected cells by IF staining MDCK cells treated with Et₃N and infected with A influenza A H1N1 (A/PR/8/34 and A/WSN/33) or H3N2 (A/Aichi/2/68) viruses. The number of influenza A NP-positive Et₃N-treated

and A/PR/8/34, A/WSN/33, or A/Aichi/2/68 virus-infected cells was dose-dependently reduced than Water-treated cells (Fig 2A).

Treatment with $Et_3N$ or baku significantly reduced the percentage of influenza A NP-positive cells infected with A/PR/8/34 (125–1000 µM $Et_3N$ or 10 µM baku; ***$P<0.001$ in Fig 2B), A/WSN/33 (250–1000 µM $Et_3N$ and 10 µM baku; **$P<0.01$ and ***$P<0.001$ in Fig 2C), or A/Aichi/2/68 (250–1000 µM $Et_3N$; **$P<0.01$ and ***$P<0.001$ in Fig 2D) viruses relative to Water-treated cells. These results indicate that $Et_3N$ inhibits influenza A H1N1 and H3N2 viral infections.

## $Et_3N$ suppressed influenza A viral NP expression and viral gene and antiviral host gene upregulation in infected MDCK cells

Next, we analyzed the expression of influenza A viral NP in MDCK cells treated with $Et_3N$ and infected with influenza A viruses using WB. Influenza A NP expression was significantly reduced in MDCK cells treated with $Et_3N$ or baku and infected with A/PR/8/34 (500–1000 µM $Et_3N$ and 10 µM baku; ***$P<0.001$ in Fig 3A), A/WSN/33 (500–1000 µM $Et_3N$ and 10 µM baku; **$P<0.01$ or ***$P<0.001$ in Fig 3B), or A/Aichi/2/68 (500–1000 µM $Et_3N$; ***$P<0.001$ in Fig 3C) viruses than in Water-treated cells.

Based on the above findings, we evaluated the SI of $Et_3N$ against influenza A virus infection. The $CC_{50}$ of $Et_3N$ in MDCK cells was determined from the WST-8 assay results shown in Fig 1B. The $IC_{50}$ values for the A/PR/8/34, A/WSN/33, and A/Aichi/2/68 strains were calculated based on the IF staining data and WB analysis results presented in Figs 2 and 3A–C. Using $IC_{50}$ values derived from IF staining, the SI values of $Et_3N$ were 56 for A/PR/8/34, 36 for A/WSN/33, and 36 for A/Aichi/2/68 (Table 1, upper section). When calculated using $IC_{50}$ values from WB analysis, the corresponding SI values were 15, 10, and 10, respectively (Table 1, lower section). These SI values, ranging from approximately 10 to over 50.

We also analyzed the expression of influenza A viral genes using RT-qPCR in $Et_3N$-treated MDCK cells and either without virus (−) or infected with influenza A virus. The relative expression levels of viral genes for *NP*, nonstructural protein 1 (*NS1*), polymerase subunits (*PA*, *PB1*, and *PB2*), matrix protein 1 (*M1*), and *M2* (Fig 3D–J) were significantly reduced in 1000 µM $Et_3N$- or 10 µM baku-treated MDCK cells infected with the virus than in Water-treated cells infected with the virus (***$P<0.001$ in Fig 3D–J).

Additionally, we analyzed the expression of interferon-β (*Ifn-β*) and myxovirus-resistant protein 1 (*Mx1*), antiviral host genes induced by viral infection [24–26]. This analysis further confirmed that $Et_3N$ inhibited influenza A viral infection. The relative expression levels of canine *IFN-β* (Fig 4A) and *Mx1* (Fig 4B) were significantly reduced in 1000 µM $Et_3N$- or 10 µM baku-treated MDCK cells infected with the virus than in Water-treated cells infected with the virus (***$P<0.001$ in Fig 4A and 4B).

These findings provide strong evidence for the inhibitory effect of $Et_3N$ against influenza A virus infection.

## $Et_3N$ reduced influenza A viral titers in the infected MDCK cell culture supernatant

We further evaluated the inhibitory effect of $Et_3N$ on influenza A virus growth in infected MDCK cells in a pre-infection experiment (Fig 5A).

We measured the viral titers of culture supernatant in A/PR/8/34-, A/WSN/33-, or A/Aichi/2/68-infected MDCK cells following treatment with $Et_3N$ (Fig 5A). The viral titers in the 2 mM $Et_3N$- or 20 µM baku-treated virus-infected MDCK cell culture supernatants (Fig 5B–D) were significantly lower than those in the culture supernatant of virus-infected cells treated with Water (***$P<0.001$ in Fig 5B, 5C, and 5D). These results indicate that $Et_3N$ inhibits influenza A H1N1 and H3N2 virus growth, suggesting that $Et_3N$ suppresses influenza A viral release and/or replication by targeting viral or host cell factors.

## $Et_3N$ did not inhibit influenza A viral NA and RdRp activities

As shown in Fig 5, $Et_3N$ treatment reduced the viral titers of culture supernatant in influenza A viruses-infected MDCK cells. This suggests that $Et_3N$ may have inhibited virion release from the host cell membrane, possibly by affecting the sialidase activity of the viral NA [27] and/or interfering with viral genome replication in the host cell nucleus through the viral RdRp [28].

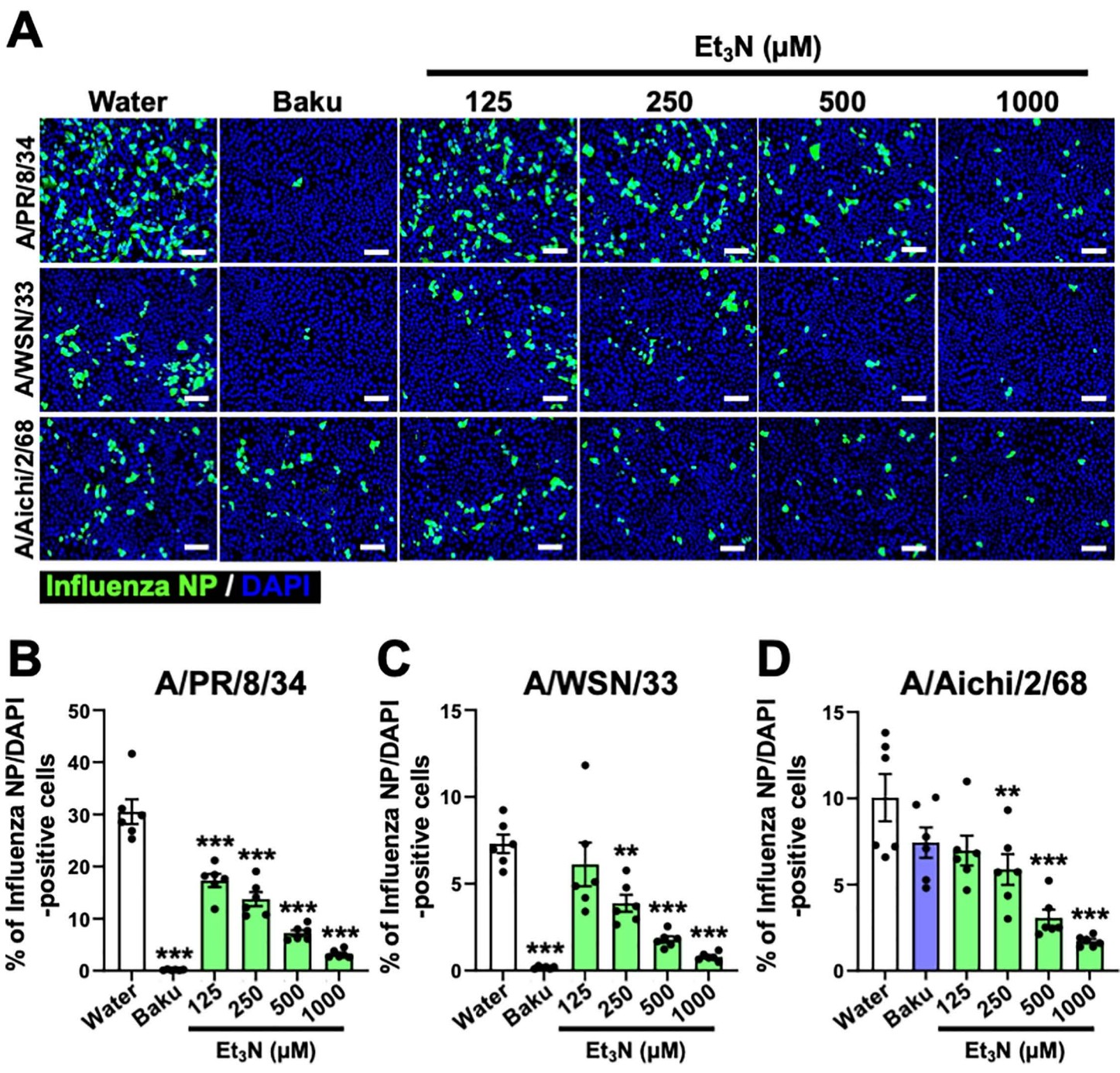

**Fig 2. Immunofluorescence analysis of the inhibitory effect of Et₃N on influenza A virus infection. (A)** IF staining for influenza NP in Et₃N-treated MDCK cells infected with influenza A H1N1 (A/PR/8/34 and A/WSN/33) or H3N2 (A/Aichi/2/68) viruses. The white scale bars in each image represent 100 μm. **(B–D)** The percentage of NP-positive cells that calculated relative to the total number of DAPI-stained nuclei. A/PR/8/34 **(B)**, A/WSN/33 **(C)**, or A/Aichi/2/68 **(D)**. Data are expressed as the mean ± SEM (n = 6) from two independent experiments. **P < 0.01 and ***P < 0.001 indicate statistical significance compared to the negative control (Water), as determined by one-way ANOVA followed by Dunnett's post-hoc test. The results were consistently reproducible across experiments.

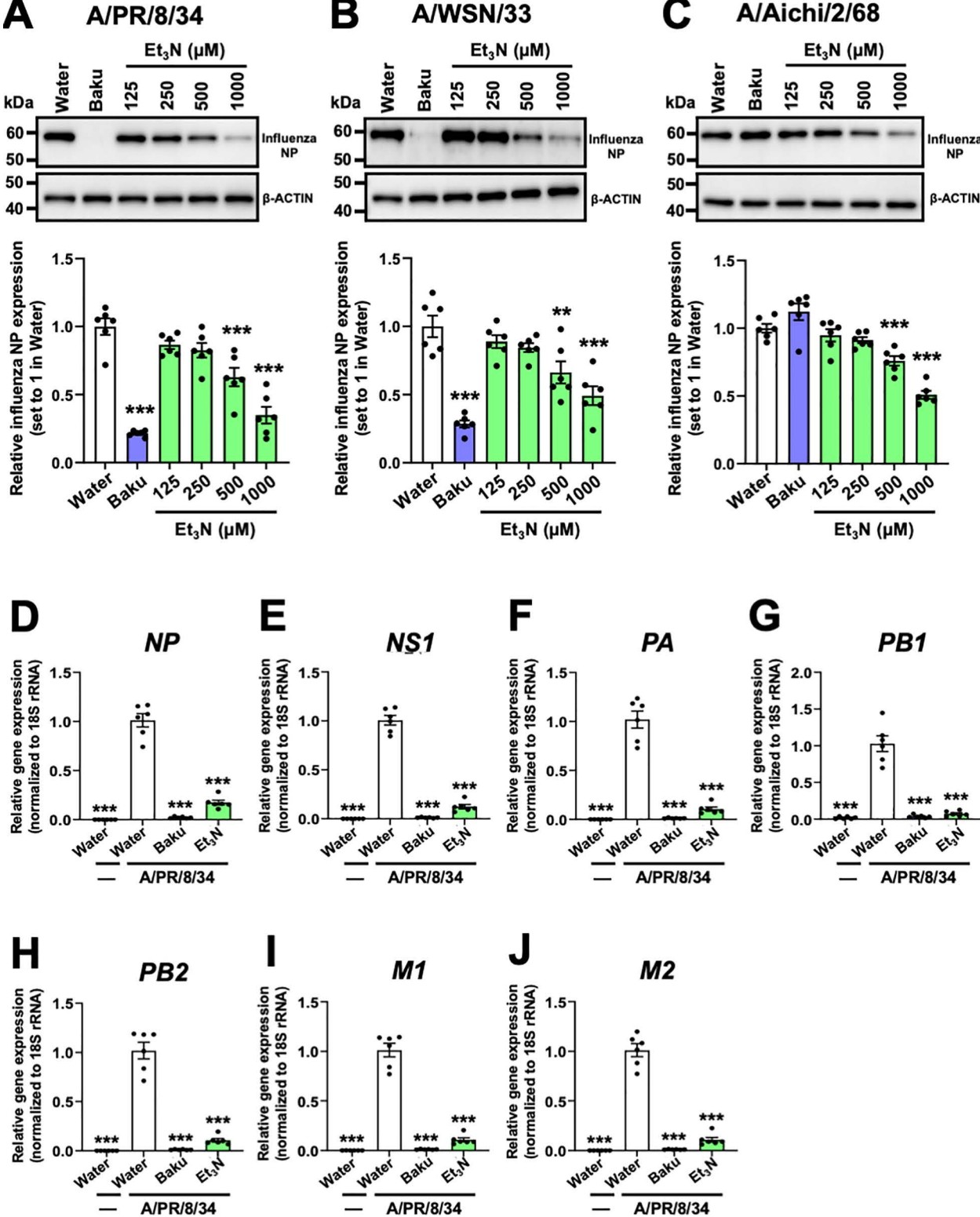

**Fig 3. Inhibitory effects of Et₃N on the expressions of influenza A viral NP and genes in virus-infected MDCK cells. (A–C)** WB analysis of influenza A viral NP expression in MDCK cells treated with Et₃N and infected with influenza A viruses. Water (control) = 1. A/PR/8/34 **(B)**, A/WSN/33 **(C)**, or

A/Aichi/2/68 **(D)**. Data are presented as the mean ± SEM (n = 6) of two independent experiments. **P < 0.01 and ***P < 0.001 indicate statistical significance compared to the negative control (Water), as determined by one-way ANOVA followed by Dunnett's post-hoc test. **(D–J)** The gene expression levels of viral *NP* **(D)**, *NS1* **(E)**, *PA* **(F)**, *PB1* **(G)**, *PB2* **(H)**, *M1* **(I)**, and *M2* **(J)** in MDCK cells treated with Et₃N and infected with A/PR/8/34 determined by RT-qPCR. Virus-infected Water-treated cells = 1. Data are presented as the mean ± SEM (n = 6) of two independent experiments. ***P < 0.001 indicates statistical significance compared to virus-infected Water-treated cells, as determined by one-way ANOVA followed by Tukey's post-hoc test. Results were consistently reproducible across experiments.

We performed an NAI assay either without virus (−) or with influenza A viral particles ([Fig 6A](–D)). Compared to Water (2%), Et₃N (2 mM) did not inhibit the sialidase activity of A/PR/8/34, A/WSN/33, or A/Aichi/2/68 ([Fig 6B](–D)) viral particles after 3, 6, and 24 h. In contrast, 0.5 μM OC strongly inhibited viral sialidase activity at all time points (***P < 0.001 in [Fig 6B](–D)). These results indicate that Et₃N does not inhibit the NA activity of influenza A H1N1 and H3N2 viruses.

Next, we performed a minigenome assay in MDCK cells expressing viral RdRp [23]. Compared with 2% Water, 2 mM Et₃N did not inhibit influenza A H1N1 viral RdRp activity in MDCK cells ([Fig 6E](5)), whereas ribavirin (25 μM) significantly

**Table 1. The selectivity index (SI) values of Et₃N against anti-influenza A virus activity.**

| Experiment | Influenza virus strain | IC₅₀ (μM) | CC₅₀ (μM) | SI values (CC₅₀/IC₅₀) |
|---|---|---|---|---|
| Infected cells (IF staining) | A/PR/8/34 | $1.8 \times 10^2$ | $1.0 \times 10^4$ | 56 |
| | A/WSN/33 | $2.8 \times 10^2$ | $1.0 \times 10^4$ | 36 |
| | A/Aichi/2/68 | $2.8 \times 10^2$ | $1.0 \times 10^4$ | 36 |
| NP expression (WB analysis) | A/PR/8/34 | $6.8 \times 10^2$ | $1.0 \times 10^4$ | 15 |
| | A/WSN/33 | $9.6 \times 10^2$ | $1.0 \times 10^4$ | 10 |
| | A/Aichi/2/68 | $1.0 \times 10^3$ | $1.0 \times 10^4$ | 10 |

Note: The SI values were calculated using IF staining (upper section) or WB analysis (lower section) of influenza NP expression in Et₃N-treated MDCK cells infected with influenza A virus strains. SI was calculated using the formula: SI = CC₅₀/ IC₅₀. CC₅₀ of Et₃N in MDCK cells was determined based on the WST-8 assay results shown in [Fig 1B]. IC₅₀ of Et₃N for A/PR/8/34, A/WSN/33, and A/Aichi/2/68 strains was calculated from IF staining or WB analysis shown in [Figs 2] and [3A](–C), respectively.

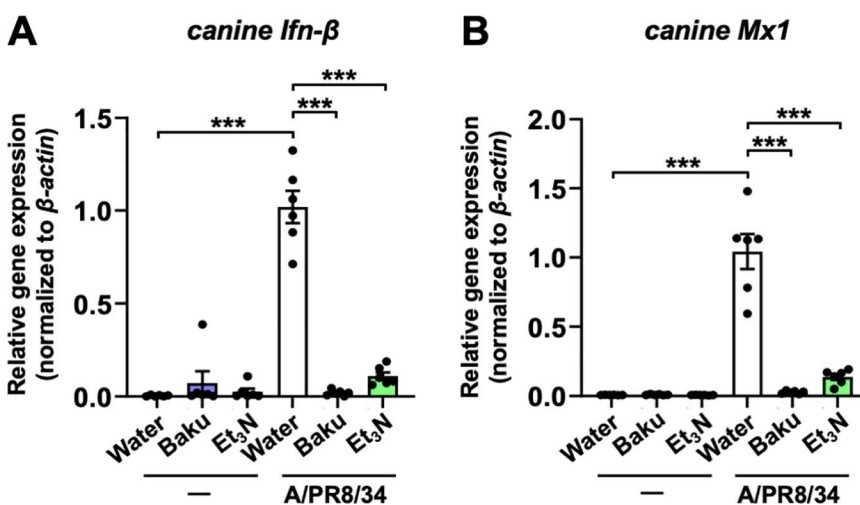

**Fig 4. Inhibitory effects of Et₃N on the upregulation of antiviral host genes in influenza A virus-infected MDCK cells.** The expression levels of canine *Ifn-β* **(A)** and *Mx1* **(B)** determined by RT-qPCR. A/PR/8/34-infected Water-treated cells = 1. Data are presented as the mean ± SEM (n = 6) of two independent experiments. ***P < 0.001 indicates statistical significance for the indicated comparisons, as determined by one-way ANOVA followed by Tukey's post-hoc test. Results were consistently reproducible in this experiment.

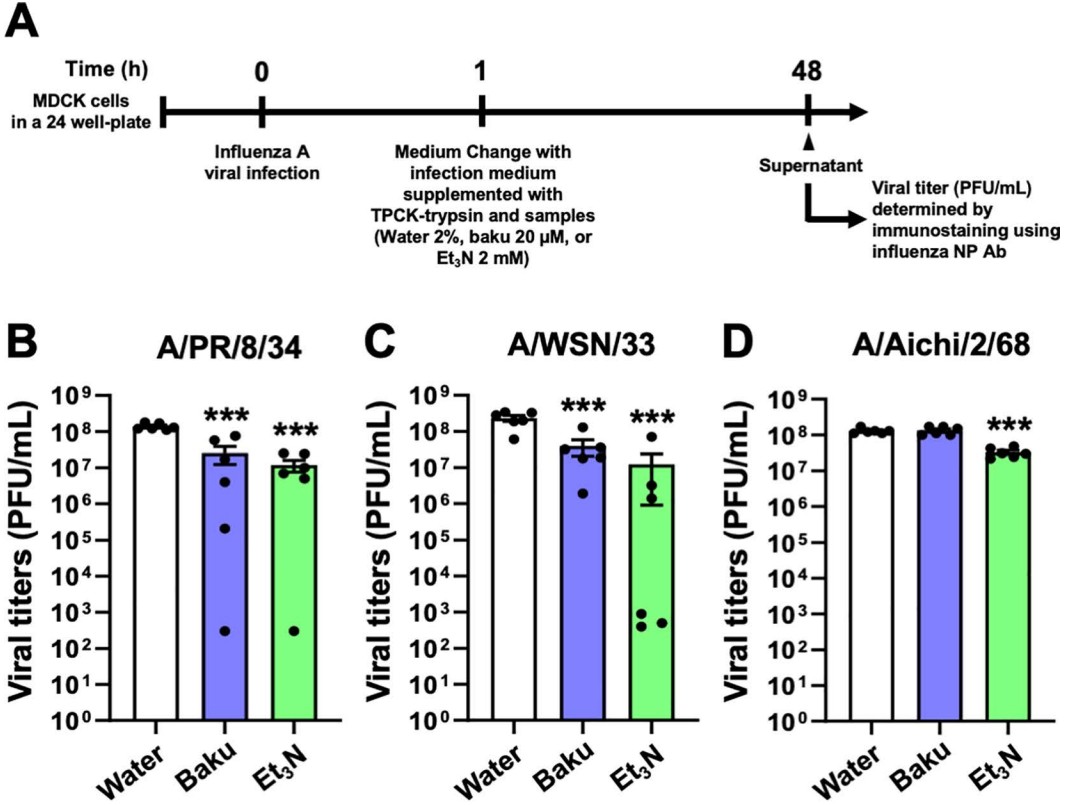

**Fig 5. Inhibitory effect of Et3N on influenza A viral growth.** (**A**) Schematic representation of the virus pre-infection experiment. (**B–D**) Influenza viral titers of culture supernatant in influenza A viruses-infected MDCK cells treated with Et3N at 48 h post-infection. A/PR/8/34 (**B**), A/WSN/33 (**C**), or A/Aichi/2/68 (**D**) viruses. Data are presented as the mean ± SEM (n = 6) of two independent experiments. ***P < 0.001 indicates statistical significance compared to the negative control (Water), as determined by one-way ANOVA followed by Dunnett's post-hoc test. Results were consistently reproducible in this experiment.

reduced RdRp activity (***P < 0.001 in Fig 6E). These results demonstrate that Et$_3$N does not inhibit the RdRp activity of influenza A H1N1 virus.

Taken together, these findings suggest that the antiviral effect of Et$_3$N is not mediated by viral NA or RdRp activity inhibition. Instead, Et$_3$N may target other viral proteins or host cell factors essential for influenza A virus growth.

## Discussion

This study demonstrated that Et$_3$N exerts antiviral activity against influenza A virus *in vitro*. Our finding provides preliminary evidence supporting the potential utility of Et$_3$N as a candidate molecule for further antiviral investigation.

Influenza virus has a negative-sense, single-stranded RNA genome consisting of eight RNA segments collectively termed viral ribonucleoproteins (vRNPs), which are individually packaged into viral particles [29]. Influenza viral proteins, including the 10 major proteins essential for viral replication (NP, NS1, NS2, PA, PB1, PB2, M1, M2, NA, and hemagglutinin [HA]), are translated from these eight vRNPs and function in the viral replication cycle within host cells [30]. During this cycle, the uncoating process begins in early endosomes that the virus enters via endocytosis [30,31]. The pH within the endosomal lumen decreases because of vacuolar adenosine triphosphatase (v-ATPase) activity, reaching pH 6.0–6.5. This decrease in pH opens the M2 channel in the viral envelope, leading to proton and potassium ion influx into the virion. In the late endosome, a further decrease in pH to 5.0–6.0 induces a conformational change in HA, facilitating fusion between the

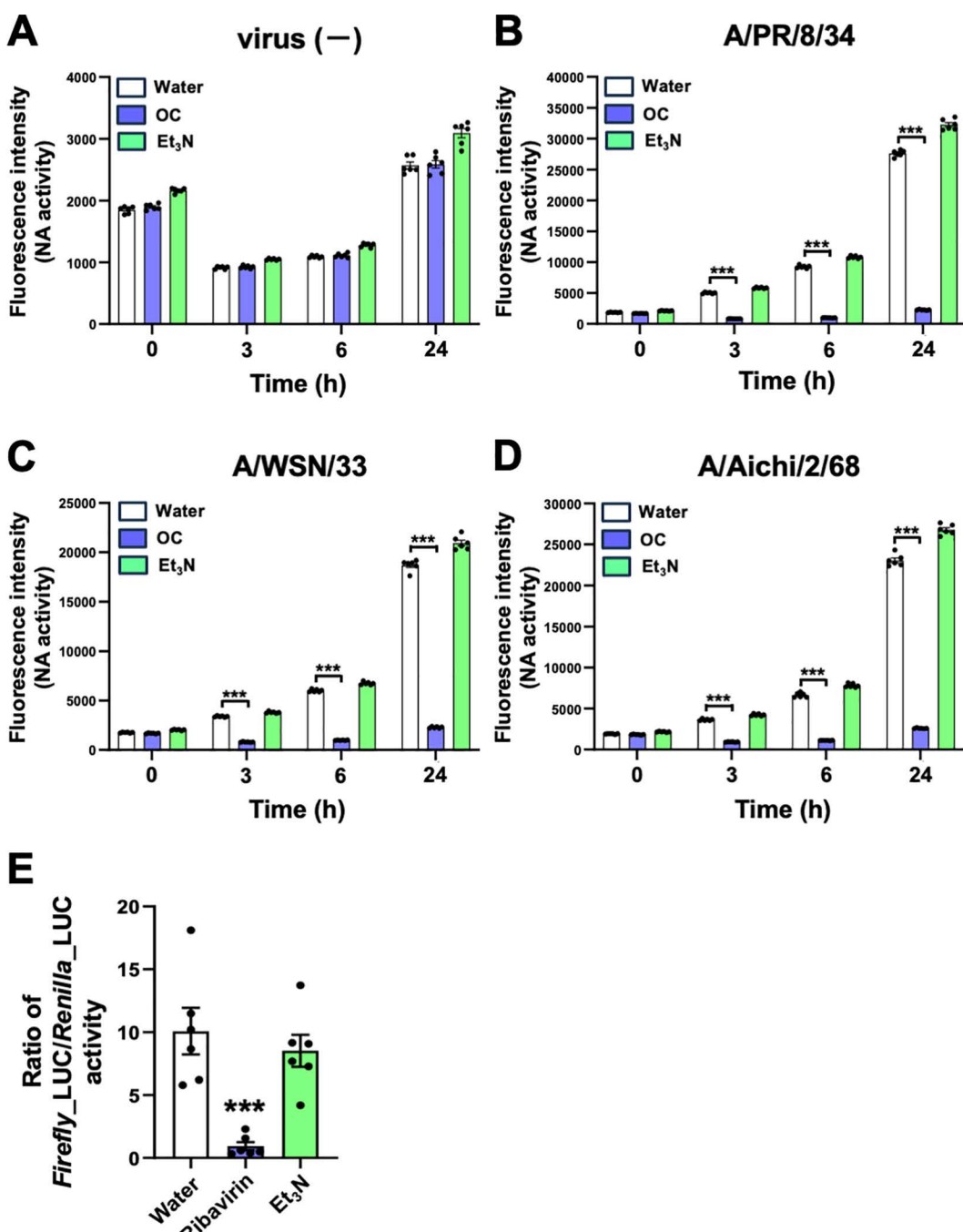

**Fig 6. Et₃N did not inhibit NA and RdRp activities. (A–D)** NAI assay using influenza A viral particles. without virus (−) **(A)**. A/PR/8/34 **(B)**, A/WSN/33 **(C)**, or A/Aichi/2/68 **(D)**. The fluorescence intensity in wells was measured at 365 nm excitation wavelength and 445 nm emission wavelength at 0, 3, 6, and 24 **h.** Data are presented as the mean ± SEM (n = 6) of two independent experiments. ***P < 0.001 indicates statistical significance for the indicated comparisons, as determined by one-way ANOVA followed by Tukey's post-hoc test. **(E)** A minigenome assay for influenza viral RdRp activity. *Firefly_*LUC and *Renilla_*LUC activities in the transfected MDCK cells were measured after 24 **h.** The RLU ratio of *Firefly_*LUC to *Renilla_*LUC was calculated. Data are presented as the mean ± SEM (n = 6) of two independent experiments. ***P < 0.001 indicates statistical significance compared to Water treatment, as determined by one-way ANOVA followed by Tukey's post-hoc test. Results were consistently reproducible in these experiments.

endosomal membrane and viral envelope, ultimately releasing vRNPs into the cytosol (uncoating). The vRNPs are then transported to the nucleus where genome replication and transcription occur. Thus, the viral uncoating process depends on a decrease in pH, which is mediated by proton influx through both endosomal v-ATPase and the viral M2 ion channel. $Et_3N$ is a typical Lewis base and strong proton ($H^+$) acceptor. We found that $Et_3N$ inhibited H1N1 and H3N2 influenza A virus infections when applied to MDCK cells together with the virus (Fig 1–4). $Et_3N$ also inhibited influenza A virus growth in virus-infected MDCK cells (Fig 5). This compound did not inhibit influenza A virus NA or RdRp activity, which are involved in influenza A virus release and/or replication in host cells (Fig 6). Therefore, 1–2 mM $Et_3N$ may suppress the decrease in pH required for viral uncoating in virus-infected MDCK cells, thus inhibiting virus infection and growth.

Previous studies have shown that $Et_3N$ inhibits sulfotransferase activity [11,12] and protein degradation [13]. Influenza A virus infection activates cholesterol sulfotransferase in mouse lungs [32], and sulfotransferase overexpression in MDCK cells increases influenza A H1N1 virus (A/Hokkaido/11/2002) production by 70- and 13-fold at 48 and 72 hpi, respectively [33]. Additionally, influenza virus infection promotes the degradation of many host cellular proteins, contributing to efficient viral propagation [34]. Therefore, $Et_3N$-mediated inhibition of sulfotransferase activity and protein degradation may play a role in attenuating influenza virus infection and growth. However, the specific molecular targets of $Et_3N$ involved in the anti-influenza A virus activity remain to be fully elucidated, necessitating further research.

### Research limitations

This study demonstrated that $Et_3N$ enhances the survival of influenza A virus-infected MDCK cells and inhibits viral infection and growth *in vitro* via mechanisms independent of viral NA and RdRp. However, several limitations should be acknowledged.

All experiments were conducted exclusively *in vitro* using MDCK cells, and the antiviral efficacy of $Et_3N$ *in vivo* remains unconfirmed. Although the SI values of $Et_3N$ against influenza A virus ranged from approximately 10 to over 50, its safety profile, toxicity, bioavailability and pharmacokinetic properties have not yet been evaluated in animal models. Furthermore, as shown in Table 1, the $IC_{50}$ values of $Et_3N$ ranged from approximately 178 to over 1,000 μM *in vitro*, which are relatively high and may not be achievable or safe at pharmacologically relevant concentrations *in vivo*. Akesson et al. reported that the plasma half-life of $Et_3N$ in humans is nearly three hours, suggesting that its bioavailability is not negligible [14]. Therefore, further structural optimization would be required to improve its antiviral potency and reduce cytotoxicity for potential *in vivo* applications.

This study did not include a direct comparison with standard antiviral drugs such as oseltamivir or baloxavir. Although $Et_3N$ exhibited antiviral activity against influenza A virus *in vitro*, the observed $IC_{50}$ values (178–>1,000 μM) are markedly higher than those of clinically approved antivirals. These relatively high $IC_{50}$ values indicate that $Et_3N$ is considerably less potent under the tested conditions. However, the aim of this study was not to present $Et_3N$ as a substitute for existing antiviral agents, but rather to explore its previously unrecognized antiviral activity. This may suggest a novel mechanism of action and serve as a foundation for future compound optimization.

The molecular mechanisms underlying the antiviral activity of $Et_3N$ remain unclear. While $Et_3N$ did not inhibit NA or RdRp activity, its precise targets—whether viral components or host cellular factors—are still unknown. Further mechanistic studies are necessary to determine whether $Et_3N$ interferes with viral entry, intracellular trafficking, host signaling pathways, or membrane dynamics involved in viral replication and release.

The antiviral effects of $Et_3N$ were demonstrated only in MDCK cells. The reproducibility of these findings in human-derived systems, such as primary bronchial epithelial cells or lung organoids, has not yet been validated. Such studies would provide more physiologically relevant insights into the potential therapeutic application of $Et_3N$ in humans.

Finally, although this study provides compelling *in vitro* evidence for the antiviral activity of $Et_3N$ against influenza A virus, its effects on other respiratory viruses—such as respiratory syncytial virus, coronaviruses (including SARS-CoV-2), rhinoviruses, adenoviruses, and parainfluenza viruses—remain unknown. Evaluating its antiviral activity against

a broader spectrum of respiratory pathogens will be essential to determine the potential of $Et_3N$ as a broad-spectrum antiviral agent.

## Conclusion

This study demonstrates that $Et_3N$ enhances the survival of infected MDCK cells, and inhibits the infection and replication of influenza A H1N1 and H3N2 viruses *in vitro*. Notably, $Et_3N$ exerts its antiviral effects through mechanisms independent of NA and RdRp, suggesting involvement of alternative viral or host cellular targets. Although the observed antiviral activity of $Et_3N$ provides novel insights, its relatively high $IC_{50}$ values and lack of *in vivo* data limit its immediate therapeutic potential. Further structural optimization will be essential to improve antiviral potency and reduce cytotoxicity. Additionally, the absence of comparative analysis with existing antivirals and the exclusive use of MDCK cells highlights the need for future investigations employing human-derived cell lines and broader respiratory viral panels. Overall, our findings suggest that $Et_3N$ serves as a novel chemical scaffold with anti-influenza potential, laying the groundwork for future development of antiviral agents that act through non-classical mechanisms.

## Supporting information

**S1 Table. Primer sequences for real-time qPCR.**
(XLSX)

**S1 Data. Raw images.**
(PDF)

## Author contributions

**Conceptualization:** Masaki Shoji, Takashi Kuzuhara.

**Formal analysis:** Masaki Shoji.

**Funding acquisition:** Masaki Shoji.

**Investigation:** Masaki Shoji, Kensuke Nakaoka, Momiji Ishikawa.

**Project administration:** Takashi Kuzuhara.

**Resources:** Yusuke Kasai, Tomoyuki Esumi, Etsuhisa Takahashi, Hiroshi Imawaga.

**Software:** Hiroshi Kido.

**Supervision:** Takashi Kuzuhara.

**Visualization:** Masaki Shoji.

**Writing – original draft:** Masaki Shoji.

**Writing – review & editing:** Takashi Kuzuhara.

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
