## [Decision Letter · Decision Letter 0]

27 Jun 2025

PONE-D-25-29309Anti-influenza activity of triethylaminePLOS ONE

Dear Dr. Kuzuhara,

Thank you for submitting your manuscript to PLOS ONE. After careful consideration, we feel that it has merit but does not fully meet PLOS ONE’s publication criteria as it currently stands. Therefore, we invite you to submit a revised version of the manuscript that addresses the points raised during the review process.

We look forward to receiving your revised manuscript.

Kind regards,

James Guevara Pulido, PhD

Academic Editor

PLOS ONE

Journal Requirements:

https://doi.org/10.1371/journal.pone.0244885

In your revision ensure you cite all your sources (including your own works), and quote or rephrase any duplicated text outside the methods section. Further consideration is dependent on these concerns being addressed.

Additional Editor Comments :

The work is engaging but should include details about the research limitations in both the manuscript's body and conclusions. It would also be beneficial to describe the selectivity indices of triethyl amine in vitro and discuss how, as a small and highly reactive molecule, triethyl amine is likely to have negligible bioavailability in vivo. This addresses the authors' challenge in accounting for this discrepancy.

Reviewers' comments:

Reviewer's Responses to Questions

**Comments to the Author**

1. Is the manuscript technically sound, and do the data support the conclusions?

Reviewer #1: Yes

Reviewer #2: Yes

2. Has the statistical analysis been performed appropriately and rigorously? 

Reviewer #1: Yes

Reviewer #2: Yes

3. Have the authors made all data underlying the findings in their manuscript fully available?

Reviewer #1: Yes

Reviewer #2: Yes

4. Is the manuscript presented in an intelligible fashion and written in standard English?

Reviewer #1: Yes

Reviewer #2: Yes

5. Review Comments to the Author

Reviewer #1: Why the authors did not study the effect of Et3N in vivo?

The rationale about thinking about testing ET3N against influenza virus is not clear.

Please check this phrase in the abstract "We investigated the antiinfluenza

activity of Et3N and found that it increased the viability of influenza A H1N1" does "increased the viability" is right"?

Reviewer #2: Specific comments:

This study demonstrated that triethylamine (Et3N), a commonly used industrial compound, exerts antiviral activity against influenza A H1N1 and H3N2 viruses in MDCK cells. At non-toxic concentrations, Et3N suppressed viral infection, reduced viral gene expression, and inhibited virus release and/or replication, although it did not target neuraminidase or RNA polymerase directly. Even if the current study offers a significant contributions influenza A virus treatment, it has numerous limitations that should be taken into account, including the following implications:

Limitations:

1. The study was conducted only in vitro using MDCK cells; no in vivo or clinical validation was performed.

2. The exact molecular target of Et3N remains unidentified, limiting mechanistic insight.

3. Potential cytotoxicity and safety in humans, especially at therapeutic doses, were not assessed.

4. The broad-spectrum antiviral potential of Et3N (against other viruses) was not evaluated.

5. Lack of comparison with standard antivirals limits understanding of its relative efficacy.

So, it is highly recommended to have a paragraph to discuss the major limitations of the present study before the section of the Conclusion.

Minor comments:

1. No line numbers.

2. In 2nd paragraph in the methods for MDCK cells, please mention how much you add from pen&strep mix to the cells and the same shoud be doe for the following paragrapg with P/S

3. Please add key words at the end of abstract.

4. In the title of figure 2 legend, please define IF and replace it with immunoflourscence for clarity.

5. The title is so limited and doesn’t express the goal of your study, so please use a representative title explianing your study brifley in a few words.

6. PLOS authors have the option to publish the peer review history of their article (what does this mean? ). If published, this will include your full peer review and any attached files.

**Do you want your identity to be public for this peer review?** For information about this choice, including consent withdrawal, please see our Privacy Policy .

Reviewer #1: **Yes**

Reviewer #2: No

---

## [Author Response · Author response to Decision Letter 1]

22 Jul 2025

Responses to Reviewers’ comments

Manuscript Number: PONE-D-25-29309

Masaki Shoji, Kensuke Nakaoka, Momiji Ishikawa, Yusuke Kasai, Tomoyuki Esumi, Etsuhisa Takahashi, Hiroshi Kido, Hiroshi Imawaga, Takashi Kuzuhara

“Triethylamine inhibits influenza A virus infection and growth via mechanisms independent of viral neuraminidase and RNA-dependent RNA polymerase”

Dear Professor James Guevara Pulido,

We would like to thank you and the reviewers for their positive comments on our manuscript and their constructive suggestions for improvement. As per the reviewers’ comments, we have added new experimental information and revised the manuscript accordingly.

Please find our point-by-point responses to the comments of the reviewers below.

We believe that the manuscript has been revised appropriately as per the opinions expressed, and we hope that it is now suitable for publication in PLOS ONE.

Thank you for your consideration. I look forward to hearing from you.

Sincerely yours,

Takashi Kuzuhara, Ph.D.

Professor, Laboratory of Biochemistry, Faculty of Pharmaceutical Sciences,

Tokushima Bunri University

180 Nishihamahouji, Yamashirocho, Tokushima-city, 770-8514, Japan

Tel.: +81 (88) 602-8477, Fax: +81 (88) 655-3051, E-mail: kuzuhara@ph.bunri-u.ac.jp

Editor’s comments

Additional Editor Comments:

The work is engaging but should include details about the research limitations in both the manuscript's body and conclusions. It would also be beneficial to describe the selectivity indices of triethyl amine in vitro and discuss how, as a small and highly reactive molecule, triethyl amine is likely to have negligible bioavailability in vivo. This addresses the authors' challenge in accounting for this discrepancy.

Response:

We sincerely thank the editor for the constructive and insightful feedback. In response, we have made several revisions to address the points raised.

We have added a new "Research limitations" section, placed between the Discussion and Conclusion sections (see pages 15–16, lines 427–460). This section explicitly outlines the limitations of our current study. Notably, we have now discussed the lack of in vivo data, as well as concerns related to the bioavailability and pharmacokinetic properties of triethylamine (Et₃N).

Next, we have evaluated and described the selectivity index (SI) of Et₃N against influenza A virus infection, as shown in Table 1 of the Results section (see page 11, lines 300–314). The SI was calculated using the method described in the Materials and methods section (see page 6, lines 166–174), based on CC₅₀ values determined in MDCK cells by the WST-8 assay (Fig. 1B) and IC₅₀ values derived from immunofluorescence and Western blot analyses for A/PR/8/34, A/WSN/33, and A/Aichi/2/68 strains (Figs. 2 and 3A–C). The calculated SI values ranged from approximately 10 to over 50, indicating a moderate to high degree of selectivity in vitro.

Furthermore, in the newly added Research limitations section, we acknowledge that no in vivo assessments of Et₃N’s safety, toxicity, or pharmacokinetics have been conducted. We note that the observed IC₅₀ values of Et₃N ranged from approximately 178 to over 1,000 µM in vitro, which are relatively high and may not be achievable or safe at pharmacologically relevant concentrations in vivo. Åkesson et al. (Åkesson et al., 1989) reported that the plasma half-life of Et₃N in humans is nearly three hours, suggesting that its bioavailability is not negligible. We now emphasize that further chemical optimization would be essential to improve its drug-like properties.

Lastly, the Conclusion section has been revised accordingly to reflect these points and ensure alignment with the Research limitations section.

Reviewer #1:

Comment 1:

-Why the authors did not study the effect of Et3N in vivo?

Response 1:

We appreciate the reviewer’s insightful question regarding the lack of in vivo evaluation of Et₃N. In response, we have now included a detailed discussion in the newly added "Research limitations" section of the revised manuscript, and we have updated several other sections accordingly.

Although the selectivity index (SI) values of Et₃N against influenza A virus infection ranged from approximately 10 to over 50, its safety profile, toxicity, and pharmacokinetic properties have not yet been evaluated in animal models. As shown in Table 1, the in vitro IC₅₀ values of Et₃N ranged from approximately 178 to over 1,000 µM, which are relatively high and may not be achievable or safe at pharmacologically relevant concentrations in vivo settings. Moreover, due to its small molecular size and high reactivity, Et₃N is likely to exhibit low bioavailability in vivo. This limitation is now clearly acknowledged and discussed as a key challenge that must be addressed through further structural optimization.

To address this point comprehensively, we have incorporated relevant descriptions and data in the following sections of the revised manuscript:

Abstract (page 2, lines 38–39):

“Selectivity index values of Et₃N against influenza A virus infection, ranging from approximately 10 to over 50.”

Materials and methods (pages 6–7, lines 171–179):

We have added a subsection describing the method used for SI calculation:

“The selectivity index (SI) calculation...”

(Details of the formula, CC₅₀ and IC₅₀ values, and software used are provided.)

Results (page 11, lines 304–318):

We now describe how SI values were calculated from WST-8, IF staining, and WB results (Table 1). The SI values of Et₃N were:

From IF staining: 56 (A/PR/8/34), 36 (A/WSN/33), and 36 (A/Aichi/2/68)

From WB analysis: 15, 10, and 10, respectively

These values demonstrate moderate antiviral selectivity in vitro.

Research limitations (pages 15–16, lines 431–466):

We emphasize that all experiments were conducted solely in vitro using MDCK cells, and that further in vivo studies are essential for assessing the therapeutic potential of Et₃N. The physicochemical characteristics of Et₃N likely contribute to low systemic exposure in vivo, thereby limiting its translational feasibility.

Conclusion (pages 16–17, lines 468–479):

The conclusion section now reflects these limitations and suggests future directions. Specifically, we state that while Et₃N displays promising in vitro antiviral activity, its relatively high IC₅₀ values, and absence of in vivo data highlight the need for chemical optimization and further validation in physiologically relevant models.

Comment 2:

-The rationale about thinking about testing ET3N against influenza virus is not clear.

Response 2:

We thank the reviewer for this important comment regarding the rationale for investigating the anti-influenza activity of Et₃N. Et₃N is a small, basic organic compound commonly used as a proton (H⁺) acceptor in a wide range of industrial organic synthesis processes, including the manufacture of pharmaceuticals, agrochemicals, and polymers. While high concentrations of Et₃N are known to be toxic—particularly via inhalation, which can cause respiratory tract and lung damage—its potential biological and pharmacological effects at non-toxic concentrations have not been well explored, especially in the context of infectious diseases of the respiratory tract. Given the compound’s known reactivity and membrane-penetrating properties, we hypothesized that Et₃N may exert modulatory effects on virus–host interactions in epithelial cells. We thus speculated that Et₃N might influence influenza A virus infection and/or growth efficiency. Our study revealed that Et₃N enhances the survival of influenza A virus-infected MDCK cells and inhibits viral infection and growth in vitro, through mechanisms that appear to be independent of NA and RdRp functions. We have revised the Abstract and Introduction sections to clarify this rationale:

Abstract (page 2, lines 28–34):

“Triethylamine (Et3N) is a proton (H⁺) acceptor that is widely used in various industrial organic synthesis processes, including the production of pharmaceuticals, agrochemicals, and polymers. Inhalation of high Et3N concentrations can damage human respiratory tract and lungs. Given the compound’s known reactivity and membrane-penetrating properties, we hypothesized that non-toxic concentrations of Et₃N may exert modulatory effects on virus–host interactions in epithelial cells. We thus investigated the anti-influenza activity of Et3N and…”

Introduction (page 3, lines 74–76):

“Although Et3N has been investigated in various contexts, its biological and pharmacological effects of antiviral activity at non-toxic concentrations on infectious diseases of the respiratory tract have not been extensively studied. Given the compound’s known reactivity and membrane-penetrating properties, we hypothesized that Et₃N may exert modulatory effects on virus–host interactions in epithelial cells. We thus speculated that Et₃N might influence influenza A virus infection and/or growth efficiency.”

Comment 3:

-Please check this phrase in the abstract "We investigated the antiinfluenza

activity of Et3N and found that it increased the viability of influenza A H1N1" does "increased the viability" is right"?

Response 3:

We thank the reviewer for pointing out this phrasing issue. We agree that the original phrase "increased the viability of influenza A H1N1" was ambiguous and could be misinterpreted as implying enhanced viral viability. Our intended meaning was that Et₃N improved the viability of host cells infected with influenza A virus. To clarify this, we have revised the phrasing from "increased" to "enhanced" in both the Abstract and Results sections as follows:

Abstract (page 2, line 34):

“…and found that it enhanced the…”

Results (page 8, line 235):

“Et3N enhanced influenza A H1N1 and H3N2 virus-infected MDCK…”

Reviewer #2:

Specific comments:

This study demonstrated that triethylamine (Et3N), a commonly used industrial compound, exerts antiviral activity against influenza A H1N1 and H3N2 viruses in MDCK cells. At non-toxic concentrations, Et3N suppressed viral infection, reduced viral gene expression, and inhibited virus release and/or replication, although it did not target neuraminidase or RNA polymerase directly. Even if the current study offers a significant contributions influenza A virus treatment, it has numerous limitations that should be taken into account, including the following implications:

Response:

We thank Reviewer#2 for their thoughtful suggestions and insights, which have enriched the manuscript and produced a better and more balanced account of the research.

Limitations:

Comment 1:

1. The study was conducted only in vitro using MDCK cells; no in vivo or clinical validation was performed.

Response 1:

We appreciate the reviewer’s insightful question regarding the lack of in vivo evaluation of Et₃N. In response, we have now included a detailed discussion in the newly added "Research limitations" section of the revised manuscript, and we have updated several other sections accordingly.

Although the selectivity index (SI) values of Et₃N against influenza A virus infection ranged from approximately 10 to over 50, its safety profile, toxicity, and pharmacokinetic properties have not yet been evaluated in animal models. As shown in Table 1, the in vitro IC₅₀ values of Et₃N ranged from approximately 178 to over 1,000 µM, which are relatively high and may not be achievable or safe at pharmacologically relevant concentrations in vivo settings. This limitation is now clearly acknowledged and discussed as a key challenge that must be addressed through further structural optimization.

To address this point comprehensively, we have incorporated relevant descriptions and data in the following sections of the revised manuscript:

Abstract (page 2, lines 38–39):

“Selectivity index values of Et₃N against influenza A virus infection, ranging from approximately 10 to over 50.”

Materials and methods (pages 6–7, lines 171–179):

We have added a subsection describing the method used for SI calculation:

“The selectivity index (SI) calculation...”

(Details of the formula, CC₅₀ and IC₅₀ values, and software used are provided.)

Results (page 11, lines 304–318):

We now describe how SI values were calculated from WST-8, IF staining, and WB results (Table 1). The SI values of Et₃N were:

From IF staining: 56 (A/PR/8/34), 36 (A/WSN/33), and 35 (A/Aichi/2/68)

From WB analysis: 15, 10, and 10, respectively

These values demonstrate moderate antiviral selectivity in vitro.

Research limitations (pages 15–16, lines 431–466):

We emphasize that all experiments were conducted solely in vitro using MDCK cells, and that further in vivo studies are essential for assessing the therapeutic potential of Et₃N. The physicochemical characteristics of Et₃N likely contribute to low systemic exposure in vivo, thereby limiting its translational feasibility.

Conclusion (pages 16–17, lines 468–479):

The conclusion section now reflects these limitations and suggests future directions. Specifically, we state that while Et₃N displays promising in vitro antiviral activity, its relatively high IC₅₀ values, and absence of in vivo data highlight the need for chemical optimization and further validation in physiologically relevant models.

Comment 2:

2. The exact molecular target of Et3N remains unidentified, limiting mechanistic insight.

Response 2:

While our results demonstrate that Et₃N does not inhibit NA or RdRp activity, the precise molecular targets responsible for its antiviral effects—whether viral or host cellular components—remain unidentified. Given that Et₃N is a typical Lewis base and a strong proton (H⁺) acceptor, we hypothesize that it may interfere with endosomal acidification, a critical step required for viral uncoating during entry into host cells. By buffering intracellular pH, Et₃N may suppress the pH-dependent conformational changes necessary for viral fusion, thereby inhibiting infection and growth.

However, this remains speculative, and we fully acknowledge that additional mechanistic studies are necessary to determine whether Et₃N affects viral entry, intracellular trafficking, host signaling pathways, or membrane dynamics involved in the influenza virus life cycle. To address this point, we have added the following text to the Research limitations section of the revised manuscript:

Research limitations (pages 16, lines 452–456):

“The molecular mechanisms underlying the antiviral activity of Et₃N remain unclear. While Et₃N did not inhibit NA or RdRp activity, its precise targets—whether viral components or host cellular factors—are still unknown. Further mechanistic studies are necessary to determine whether Et₃N interferes with viral entry, intracellular trafficking, host signaling pathways, or membrane dynamics involved in viral replication and release.”

Comment 3:

3. Potential cytotoxicity and safety in humans, especially at therapeutic doses, were not assessed.

Response 3:

We thank the reviewer for this important comment regarding the cytotoxicity and safety profile of Et₃N, particularly in the context of potential human application. In response, we evaluated the SI values of Et₃N against influenza A virus infection using standard cytotoxicity and antiviral assays in MDCK cells. The SI was calculated as the ratio of the 50% cytotoxic concentration (CC₅₀) to the 50% inhibitory concentration (IC₅₀), according to the formula: SI = CC₅₀ / IC₅₀. The CC₅₀ was obtained from WST-8 assay results (Fig 1B), and IC₅₀ values were derived from both IF and WB analyses (Figs 2 and 3A–C). The SI values ranged from approximately 10 to over 50, depending on the viral strain and assay method. These values suggest a favorable in vitro therapeutic window; however, we fully acknowledge that these findings are limited to MDCK cells, which are non-human epithelial cells. To address the broader concern regarding human safety and relevance, we have added a discussion in the Research limitations section emphasizing the need for further validation in human-derived models, such as primary bronchial epithelial cells or lung organoids. These models would provide a more physiologically relevant assessment of Et₃N’s safety and antiviral efficacy. We have also included this limitation in the Concl

---

## [Editor Report · Decision Letter 1]

24 Jul 2025

Triethylamine inhibits influenza A virus infection and growth via mechanisms independent of viral neuraminidase and RNA-dependent RNA polymerase

PONE-D-25-29309R1

Dear Dr. Kuzuhara,

We’re pleased to inform you that your manuscript has been judged scientifically suitable for publication and will be formally accepted for publication once it meets all outstanding technical requirements.

Kind regards,

James Guevara Pulido, PhD

Academic Editor

PLOS ONE

Additional Editor Comments :

The authors have addressed the suggested corrections and now believe the manuscript is ready for publication.

---

## [Editor Report · Acceptance letter]

PONE-D-25-29309R1

PLOS ONE

Dear Dr. Kuzuhara,

I'm pleased to inform you that your manuscript has been deemed suitable for publication in PLOS ONE. Congratulations! Your manuscript is now being handed over to our production team.

Kind regards,

on behalf of

Dr. James Guevara Pulido

Academic Editor

PLOS ONE